# Investigation on Charge Transport in Polypropylene Film under High Electric Field by Experiments and Simulation

**Xiying Dai** [1]**, Fuqiang Tian** [2,*]**, Fei Li** [1]**, Shuting Zhang** [2]**, Zhaoliang Xing** [1,*] **and Jinbo Wu** [2]

1   State Key Laboratory of Advanced Power Transmission Technology, Beijing 102209, China; daixiying311@163.com (X.D.); lifei_geiri@163.com (F.L.)
2   School of Electrical Engineering, Beijing Jiaotong University, Beijing 100044, China; 19117036@bjtu.edu.cn (S.Z.); 20126204@bjtu.edu.cn (J.W.)
*   Correspondence: fqtian@bjtu.edu.cn (F.T.); 18846136536@163.com (Z.X.)

**Abstract:** The charge transport in polypropylene was studied under DC electric fields at different temperatures. By the experimental measurement and simulation of the BCT model, we studied conduction currents, breakdown strength, and space charge distribution. In particular, the conduction characteristics under high temperature and high field, especially the conduction characteristics before the breakdown, were studied by systematic experiments, and the conduction characteristics and the breakdown mechanism were further studied by simulation. The results show that in the process of measuring conduction currents until breakdown, both high temperature and high electric field will promote charge transport. However, the free volume will increase at high temperature, which will easily lead to faster charge transport and breakdown. In the breakdown process at different temperatures, there are different breakdown mechanisms. At 20–60 °C, the electric breakdown process has mainly occurred in polypropylene film, and the breakdown strength is almost unchanged. At 80 °C, electric breakdown and thermal breakdown act together, and the charge transport is faster, and the breakdown field becomes smaller. Finally, we conclude that thermal stress plays a very important role in charge transport. In a high-temperature environment, the volume expansion of polypropylene will promote charge transport, and the insulation of polypropylene capacitor films will be damaged.

**Keywords:** charge transport; DC breakdown; conduction current; polypropylene film

## 1. Introduction

A metallized film capacitor, insulated with biaxial-orientated polypropylene (BOPP), is widely used as the filter capacitor for electric energy converter circuits, and is also one of the key pieces of equipment in the inverter station in high-voltage direct current (HVDC) transmission systems [1]. In the process of its service in the HVDC inverter station, its insulation sustains the combined effects of high DC electric field and high temperature over the life, leading to early failure, and even breakdown of the BOPP film [2]. Compared with the insulation materials in other electrical equipment, such as XLPE in power cables, polyimide film in motors, and Nomex insulation paper in power transformers, in which the operating electric fields are normally less than 50 kV/mm, the dielectric behavior and aging process are expected to be substantially different in BOPP film regularly operating under an electric filed more than 200 kV/mm and temperatures up to 80 °C. Extra high-field dielectric behavior, and the corresponding mechanism of insulating materials, are closely related to the microscopic physical process of charge transport. However, there are few studies on the charge transport characteristics of BOPP film under an extra high electric field (200 kV/mm to breakdown electric field), due to the limitation of the experimental conditions. The pulsed electric acoustic (PEA) method is powerful in measuring space charge distribution in thick insulating films (such as polyethylene with a thickness of more

than 100 μm), while it cannot be used to study the BOPP film with a thickness of less than 10 μm, due to the space resolution limitation [3].

A conduction current varying with time, electric field or/and temperature, is a macroscopic expression of microscopic charge transport characteristics, but it cannot provide deep insight into the charge transport process. In recent years, the numeric simulation method has been developed in order to study the microscopic process of charge transport in polyethylene films and HVDC cables under electric–thermal stress. The effect of trap density and depth, charge injection barrier, temperature, carrier mobility, and other microscopic parameters, on charge transport characteristics, can be studied readily in addition to space charge formation and evolution [4–6]. The research on charge transport simulation is necessary and helpful to gain a better understanding of the macroscopic dielectric behavior in BOPP under an extra high electric field with varying temperatures.

In this study, we investigated the conduction current from 230 kV/mm (normal operating electric field in capacitors) to breakdown electric field, and the DC breakdown strength of a commercial BOPP film under different temperatures, by both experiments and the simulation method. The charge transport behavior, and related mechanism, were elucidated, which can contribute to the understanding of dielectric behavior, and evaluating the reliability of the BOPP film insulation in the HVDC filter capacitor.

## 2. Materials and Methods

### 2.1. Materials

Commercial BOPP films of highly isotactic polypropylene grades with a type of 03-9D HC300BF were used for experiments. The films were provided as film rolls with a thickness of 9 μm. The crystallinity of films is about 52%. Aluminum electrodes of 20 mm diameter were made on both sides of polypropylene film for experiments to ensure good electrical contact with the cylindrical copper electrodes connecting the power source. The BOPP films were dried at 60 °C under short-circuited conditions for 12 h and the silicone oil was dried at 60 °C for 12 h before experiments.

### 2.2. Conductivity Current Test

The conductivity of BOPP films was measured under DC electric field from 230 kV/mm to electrical breakdown field at different temperatures (20 °C, 40 °C, 60 °C, 80 °C) using the high-voltage DC power supply, the thermostatic drying oven, the Keithley 6485 electrometer and the three-electrode system.

The step-jump electric field was adopted from 230 kV/mm to electric breakdown field increasing by 40 kV/mm. The conduction current was measured for 6000 s after the voltage stabilized. The conduction current testing system is shown in Figure 1.

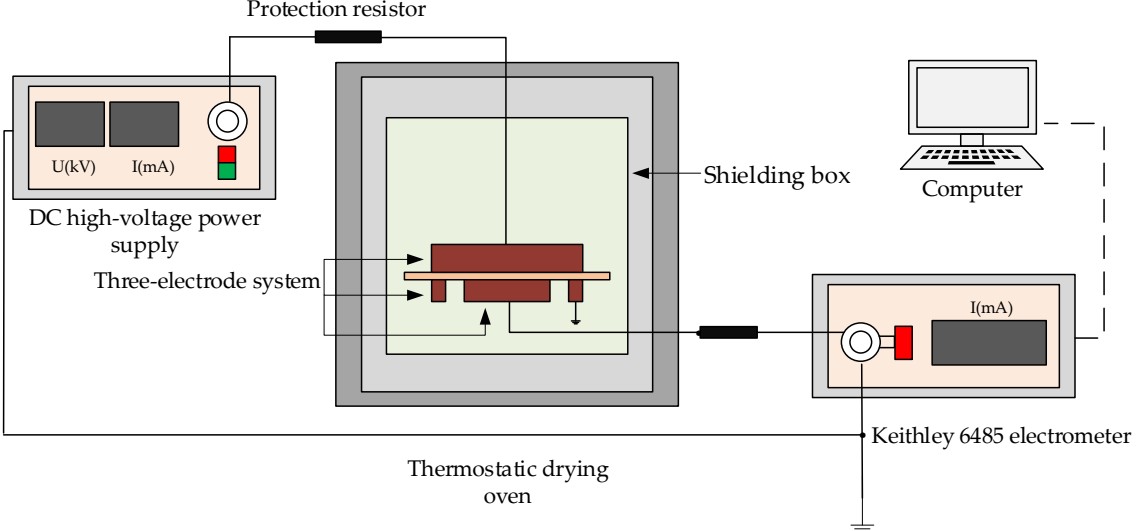

**Figure 1.** Conduction current testing system.

### 2.3. DC Breakdown Strength

DC breakdown strength was measured at different temperatures (20 °C, 40 °C, 60 °C, 80 °C) using the program controlled electric breakdown tester in the thermostatic drying oven. The temperature accuracy of the thermostatic drying oven was better than ±1.0 °C. A voltage ramp of 500 V/s was applied to the film samples until the breakdown occurred and the voltage was recorded. Ten BOPP film samples were tested to ensure the reliability of the experimental results. The DC breakdown strength was tested in dried simethicone to avoid partial discharges. Electric breakdown strength was calculated and fitted with either single two-parameter or additively mixed two-subpopulation Weibull distributions.

### 2.4. Bipolar Charge Transport Simulation Model

In this study, the bipolar charge transport model (BCT model) takes into account bipolar charge injection, transport, charge trapping, recombination and extraction processes as discussed in detail in other publications [4]. The charge transport process in the polypropylene film can be described by the following equations (conduction equation, current continuity equation and Poisson's equation):

$$J_{e/h} = \mu_{e/h} n_{e\mu/h\mu} E(r,t) = v_{e/h} n_{e\mu/h\mu},$$

(1)

$$\frac{\partial n_a(x,t)}{\partial t} + \frac{\partial j_a(x,t)}{\partial x} = s_a(x,t),$$

(2)

$$\frac{\partial^2 V(x)}{\partial x^2} = -\frac{\rho(x)}{\varepsilon_0 \varepsilon_r},$$

(3)

where $E$ is electric field, $j$ is transport current density, x is the position, $t$ is the time, $\varepsilon$ is the permittivity, $n_e$ and $n_h$ are the total electron and hole density. Further, $n_{e\mu}$ and $n_{h\mu}$ are mobile electron and hole density, and $\mu_e$ and $\mu_h$ are effective mobility of electrons and holes. $S_a$ is the source term and represents local charge density changes, including all the changes through phenomena such as trapping, detrapping and recombination processes.

Considering both the effects of electric field and temperature on the processes of carrier injection from the electrodes into the bulk insulation and carrier extraction from interface into the electrodes, the electrode current combined the field assisted thermal electron emission (Schottky emission) and FN field emission (Fowler–Nordheim tunneling) (Equation (4)).

$$J_{e/h}(0/d,t) = AT^2 \exp(-\frac{w_{ei/hi}}{kT}) \exp(\frac{e}{kT}\sqrt{\frac{eE(0/d,t)}{4\pi\varepsilon_0\varepsilon_r}}) + A_{ei/hi}E^2 \exp(-\frac{B_{ei/hi}}{E}),$$

(4)

Generally, ionic conduction dominates the conduction process under low electric field while electronic conduction is predominant under high electric field [7]. The simulation model ignored the heteropolar space charge caused by the impurity ionization in view of the high electric field for both experiments and simulations. The parameters such as injection barrier, trap depths, distance between traps, trapping coefficient, recombination coefficients, deep trap depths and deep trap densities used in the charge transport simulation are shown in Table 1, which are similar to the typical values in other publications [8–12].

**Table 1.** Parameters for BCT model.

| Model Parameter | Numerical Value | Unit |
|---|---|---|
| Electron injection barrier | 1.5 | eV |
| Hole injection barrier | 1.5 | eV |
| Electron jump barrier height | 0.72 | eV |
| Hole jump barrier height | 0.72 | eV |
| Jump distance between electronic shallow traps | $3 \times 10^{-9}$ | m |
| Jump distance between shallow hole traps | $3 \times 10^{-9}$ | m |
| Electron trapping coefficient | 0.1 | $s^{-1}$ |
| Hole trapping coefficient | 0.1 | $s^{-1}$ |
| Deep trap depth of electron | 1.4 | eV |
| Deep trap depth of hole | 1.4 | eV |
| Deep trap density of electron | $1000 \times 10^{19}$ | $m^{-3}$ |
| Deep trap density of hole | $1000 \times 10^{19}$ | $m^{-3}$ |
| Recombination coefficient $S_1$, $S_2$, $S_3$ | $1 \times 10^{-5}$ | $m^3 \cdot C^{-1} \cdot s^{-1}$ |
| Recombination coefficient $S_0$ | 0 | $m^3 \cdot C^{-1} \cdot s^{-1}$ |

## 3. Results and Discussion

### 3.1. Experimental and Simulation Results on Conduction Current and Charge Transport

Figure 2 shows the charging currents as a function of time at various electric fields (E) and different temperatures (T).

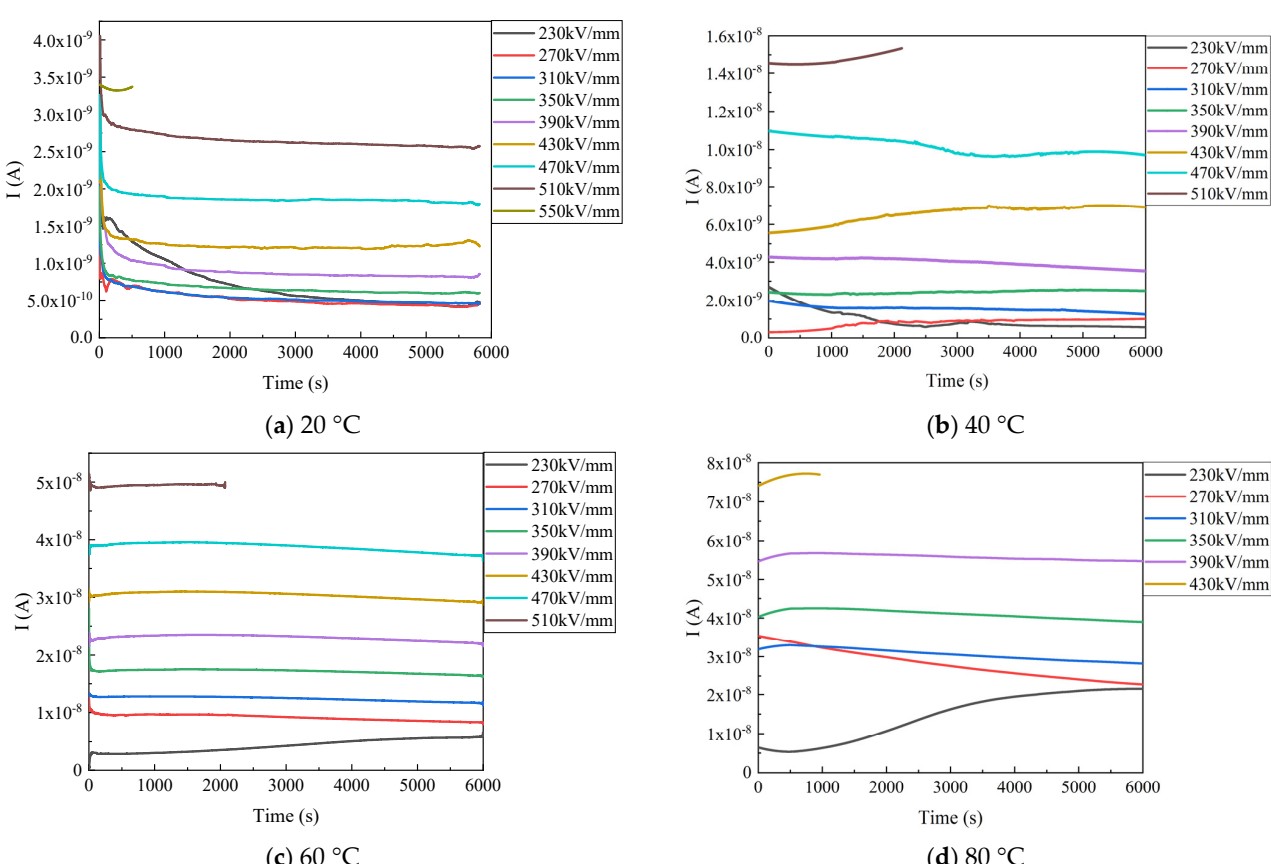

**Figure 2.** Time dependence of conduction currents in BOPP at different temperatures and various fields.

At 20 °C, the steady-state conduction current is about $4.5 \times 10^{-10}$ A and the current shows very slight dependence on the electric field when the electric field increases from 230 to 310 kV/mm. The conduction current shows a more significant increase, from $6.0 \times 10^{-10}$ A to $2.5 \times 10^{-9}$ A, for the electric field from 350 to 510 kV/mm. Then, the electrical breakdown happens under 550 kV/mm, after the electric field was applied for

about 500 s. The conduction current shows a slight increase with time under 430 kV/mm, and decays for the other electric fields.

At 40 °C, the steady-state conduction current shows a slight increase, from $6.2 \times 10^{-10}$ A to $1.4 \times 10^{-9}$ A, when the electric field increases from 230 to 310 kV/mm. The conduction current shows a much more significant increase, from $1.4 \times 10^{-9}$ A to $9.9 \times 10^{-9}$ A, for the electric field from 310 to 470 kV/mm. Then, the electrical breakdown happens under 510 kV/mm, after the electric field was applied for about 2000 s. The conduction current shows a slight increase under 270 kV/mm, and a remarkable increase under 430 kV/mm and 510 kV/mm with time. The current decays under 230 kV/mm and shows flat under the other electric fields.

At 60 °C, the steady-state conduction current shows a significant increase, from $2.9 \times 10^{-9}$ A to $6.6 \times 10^{-8}$ under 230 kV/mm with time, and a slight increase in the first 500 s under 390 to 470 kV/mm. Then, the electrical breakdown happens under 510 kV/mm, after the electric field was applied for about 2000 s. The conduction current keeps almost flat with time for the other electric field.

At 80 °C, the steady-state conduction current shows a remarkable increase, from $5.5 \times 10^{-9}$ A to $2.2 \times 10^{-8}$ A, under 230 kV/mm. The current shows a slight increase in the first 500 s, and then keeps almost flat for the rest of the time, under 310 to 430 kV/mm. The conduction current shows a nearly linear decrease with time under 270 kV/mm. The electrical breakdown happens under 430 kV/mm, after the electric field was applied for about 1000 s. The conduction current shows a significant increase ($2.9 \times 10^{-9}$ A to $6 \times 10^{-8}$ A) under 230 kV/mm with time, and a slight increase in the first 500 s under 390 to 470 kV/mm.

From the conduction current results in Figure 2, we can find that the conduction current shows a significant increase with an increase in the temperature for the same electric field. The electric field for the breakdown generally shows a considerable decrease with temperature, except for 60 °C. The conduction current shows weak dependence on the electric field, when the electric field is less than 350 kV and the temperature is lower than 60 °C. The conduction current shows an abnormal increase with time, and the phenomenon becomes more significant with an increase in the temperature under 430 kV/mm at 20 °C and 40 °C, 230 kV/mm at 60 °C and 80 °C. The conduction current increases with time before the breakdown for all the tested temperatures. The abnormal increase in the conduction current with time, in polymers, has been attributed to remarkable charge injection and accumulation in the bulk of the film [13].

Figure 3 shows the electric field dependence of the conduction current value in BOPP, at 60 s and 6000 s at various temperatures. The current values at 60 s and 6000 s are adopted considering the charge transport transient process. We can find that the conduction current increases nonlinearly with varying electric fields, as shown in Figure 3(a1,b1). The electric field dependence of the conduction current becomes more prominent with increasing temperatures. The change in trend of the conduction current is more regular at 6000 s than that at 60 s. The irregular change in the I vs. E characteristic is closely related to the abnormal conduction current behavior, as shown in Figure 2. The field dependence of the conduction current can be more clearly found by plotting lnI vs. lnE, as shown in Figure 3(a2,b2). The lnI changes nearly linearly with lnE at high electric fields with the slope of the curve higher than one, which is the typical characteristics of the space charge limited conduction current (SCLC) behavior [13,14]. This indicates that the conduction current is generally dominated by the SCLC process, i.e., by charge accumulation, and charge trapping and detrapping processes. The traps of polypropylene film mainly exist at the interface of crystal and amorphous regions. This can be further confirmed by the charge transport simulation shown later. For the conduction current at 60 s, as shown in Figure 3(a2), the slope of the lnI vs. lnE curve at high electric fields is 2.21 at 20 °C, 4.24 at 40 °C, 2.53 at 60 °C, and 2.55 at 80 °C. For the conduction current at 6000 s, as shown in Figure 3(b2), the slope of the lnI vs. lnE curve at high electric fields is 3.49 at 20 °C, 3.95 at 40 °C, 2.61 at 60 °C, and 2.39 at 80 °C. According to the SCLC theory, the slope

should be about two for the SCLC without traps or with traps fully filled, and that with a slope larger than two corresponds to the trap filling process by injected charges. The larger the slope, the more remarkable the charge trapping and trap filling process [15]. Hence, it is reasonable to infer from the curve characteristics based on SCLC theory that a few space charges accumulate at 60 s at 20 °C, and then the traps are quickly filled and the trapped charges increase significantly with time until 6000 s, corresponding to the slope increasing from 2.21 to 3.49. For 40 °C, the curve slope at 60 s is 4.24 and at 6000 s it is 3.95, indicating that the trap filling process is much more rapid, and much more trapped charges accumulate than at 20 °C. The smaller values of the curve slope at 60 °C and 80 °C, at both 60 s and 6000 s, implies a rapid transient process and less trapped charge accumulation, which can be attributed to the injected charges that migrate out due to the high bulk conduction at rising temperatures, as shown in Figure 2. As a whole, the lnI vs. lnE curve agrees well with the SCLC theory.

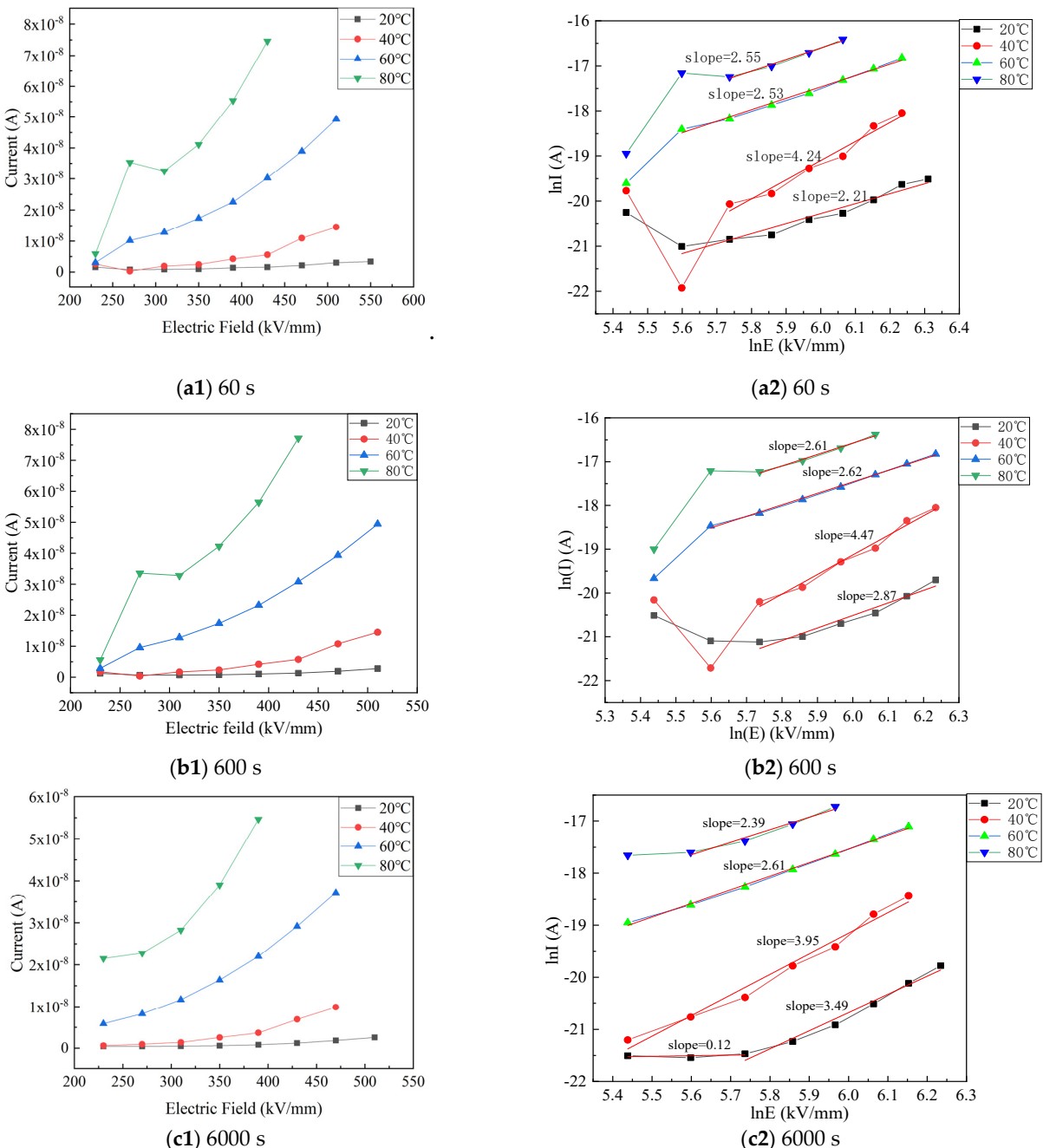

**Figure 3.** Electric field dependence of conduction current value in BOPP at 60 s, 600 s and 6000 s at various temperatures.

The lnI vs. 1/T curves are plotted to investigate the temperature dependence of the conduction current value in BOPP, at 60 s and 6000 s at various electric fields, as shown in Figure 4. We can find that the current increases significantly with increasing temperatures, and accords well with the Arrhenius relation. The slope of all the curves is about 6.5–7.5 and the activation energy is calculated according to Arrhenius relation, as shown in Table 2. As can be seen clearly, the activation energy is about 0.55–0.65 eV and shows weak electric field dependence, indicating that charge transport is mainly a thermally activated process.

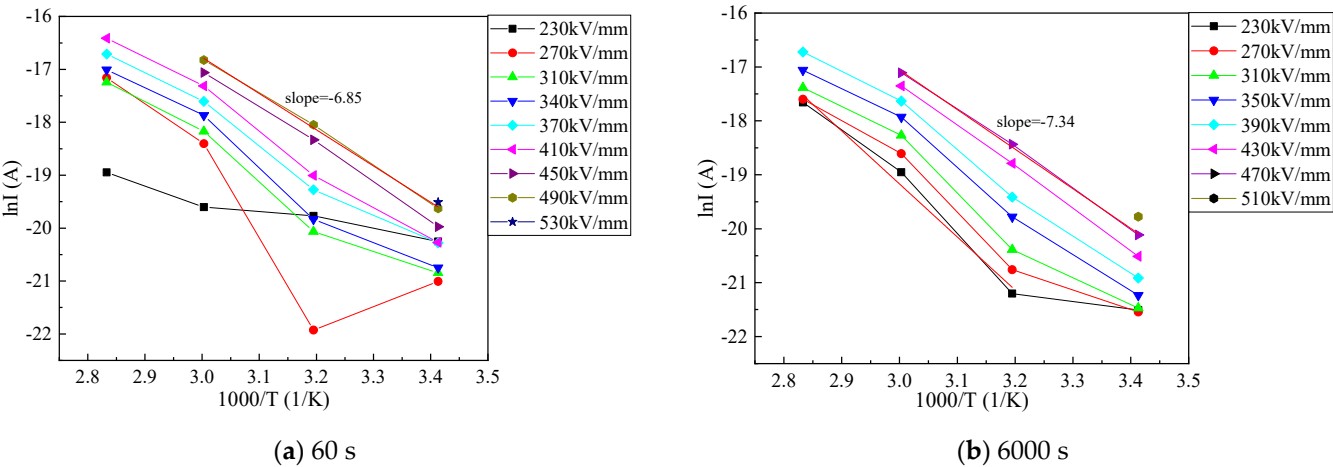

**(a)** 60 s　　　　　　　　　　　　　　　　　　　　　　　　**(b)** 6000 s

**Figure 4.** Temperature dependence of conduction current value in BOPP at 60 s and 6000 s at various electric fields.

**Table 2.** Activation energy of PPs at various electric fields.

| Electric Field (kV/mm) | 230 | 270 | 310 | 350 | 390 | 430 | 470 |
|---|---|---|---|---|---|---|---|
| 60 s | 0.66 | 0.56 | 0.64 | 0.64 | 0.60 | 0.61 | 0.59 |
| 6000 s | 0.62 | 0.64 | 0.65 | 0.64 | 0.66 | 0.63 | - |

Figures 5–8 show the simulation results of space charge distribution and electron distribution in the BOPP film at different temperatures. The electric field is applied for 6000 s and then raised directly by 40 kV/mm until breakdown. The space charge and electron distributions are plotted every 500 s at a low electric field, and once per step for the high electric field, when the charge distribution tends to reach a steady state. Both the electrons and holes are included in the space charge distribution. In order to observe the charge transport process more clearly, the total electrons, trapped electrons, and free electrons are also shown in the figures. The results show that both lots of the electrons and holes are injected from the electrodes.

Figure 5 shows that more and more electrons are injected from the cathode and move deep into the bulk of the BOPP film over time (0–6000 s), under 230 kV/mm at 20 °C. Then, the electrons around the cathode decrease, and extend deep into the bulk and toward the anode over time, under a higher electric field (>230 kV/mm). Overall, both the trapped electron and free-electron distributions become more and more uniform with time. During the transient process, the trapped electron density reaches a maximum of about $10^{21}$ m$^{-3}$ at 17,500 s, under 310 kV/mm, and the free electron density reaches a maximum of about $10^{20}$ m$^{-3}$ at 23,500 s, under 350 kV/mm. After 47,500 s, the space charge and electron distributions reach nearly steady state and show uniform distribution over the bulk of the BOPP film. The trapped electron density is about 1/10 of that of the free electrons at the steady state.

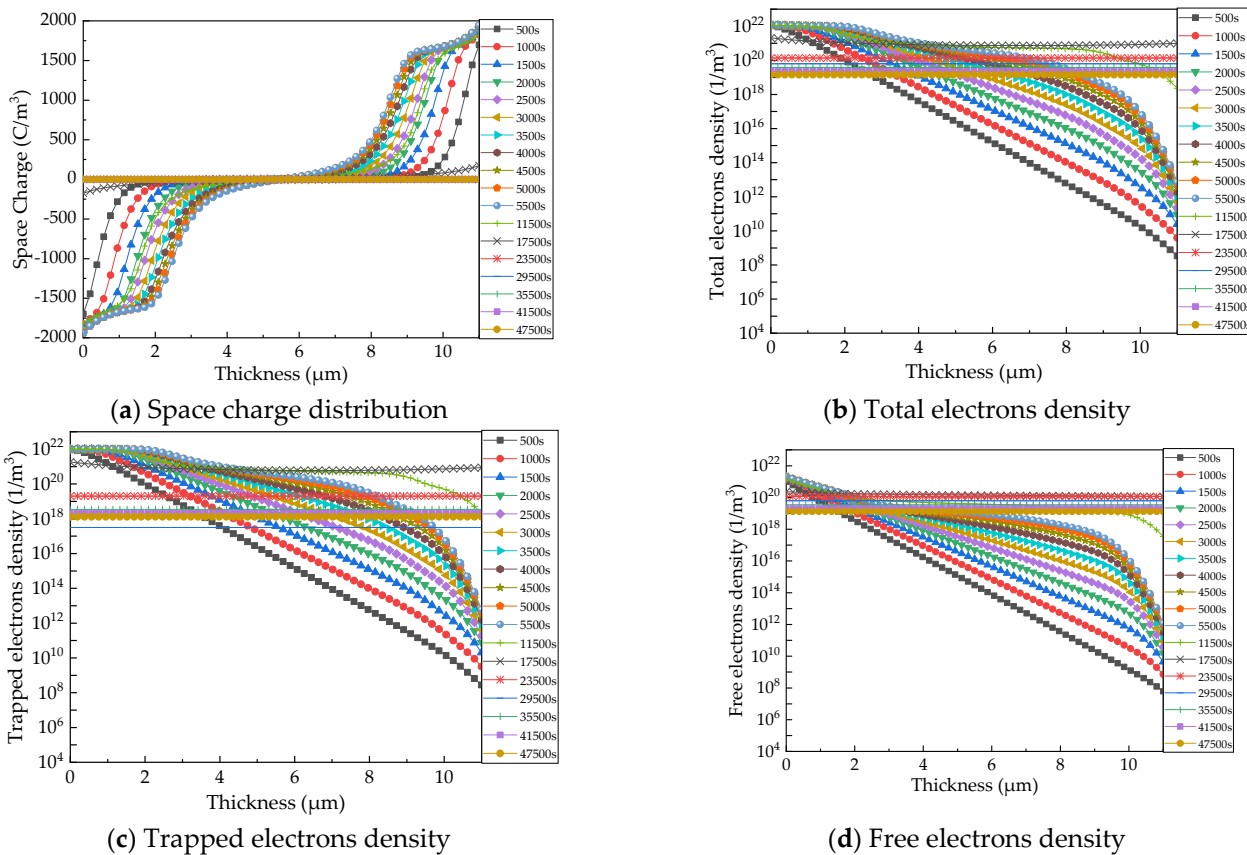

**Figure 5.** Simulation results of space charge distribution (**a**) and electrons density distribution (**b**–**d**) at 20 °C. (0 s–6000 s: 230 kV/mm; 6000 s–12000 s: 270 kV/mm; 12,000 s–18,000 s: 310 kV/mm; 18,000 s–24,000 s: 350 kV/mm; 24,000 s–30,000 s: 390 kV/mm; 30,000 s–36,000 s: 430 kV/mm; 36,000 s–42,000 s: 470 kV/mm; 42,000 s–48,000 s: 510 kV/mm).

Figure 6 shows the space charge and electron distributions at 40 °C, which exhibits a similar variation trend to that at 20 °C. The electrons initially increase over time and move towards the counter electrode. Then, the electrons tend to uniformly distribute in the bulk, at about 17,500 s. The transient process lasts for a shorter time with an increase in the temperature, as can be seen clearly from Figures 5–8. The trapped electrons firstly increase with time, and then decrease after reaching the uniform distribution.

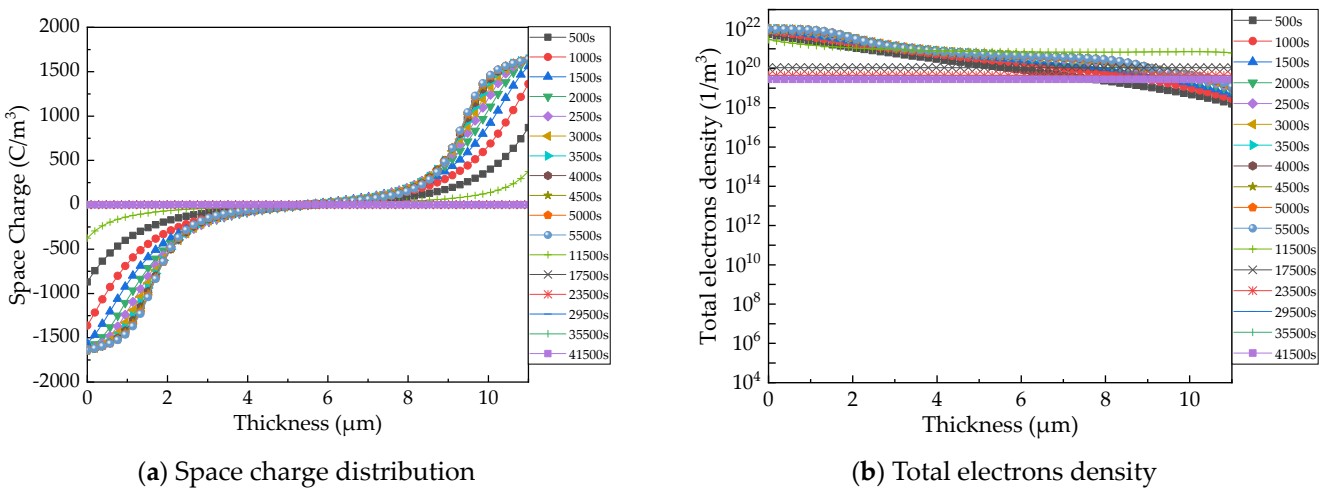

**Figure 6.** *Cont.*

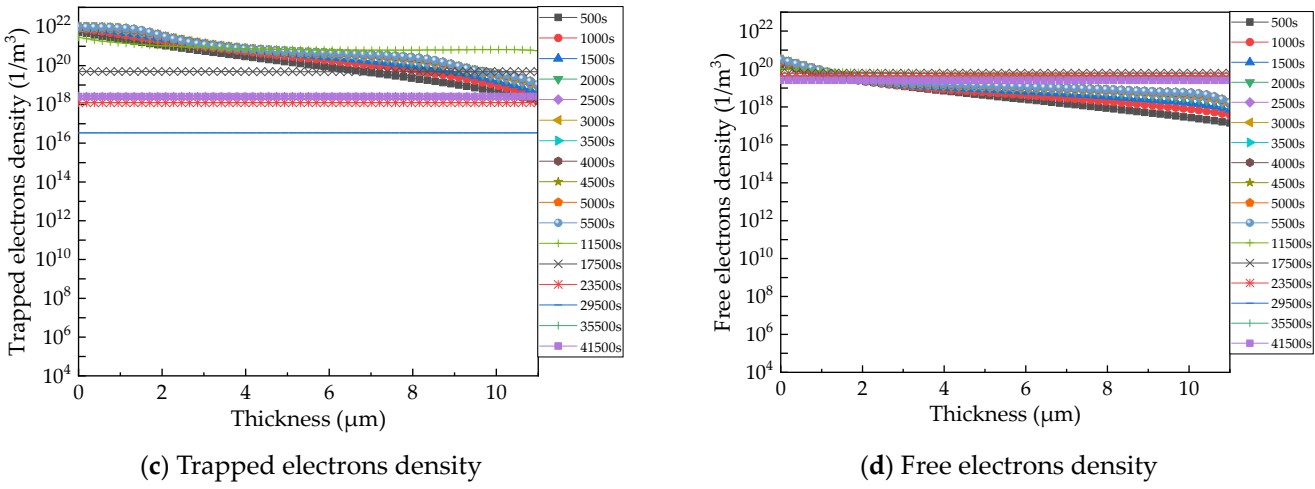

**(c)** Trapped electrons density

**(d)** Free electrons density

**Figure 6.** Simulation results of space charge distribution (**a**) and electrons density distribution (**b**–**d**) at 40 °C. (0 s–6000 s: 230 kV/mm; 6000 s–12,000 s: 270 kV/mm; 12,000 s–18,000 s: 310 kV/mm; 18,000 s–24,000 s: 350 kV/mm; 24,000 s–30,000 s: 390 kV/mm; 30,000 s–36,000 s: 430 kV/mm; 36,000 s–42,000 s: 470 kV/mm; 42,000 s–48,000 s: 510 kV/mm. When the electric field is greater than 230 kV/mm, space charge distribution and electrons density distribution are almost the same at each time).

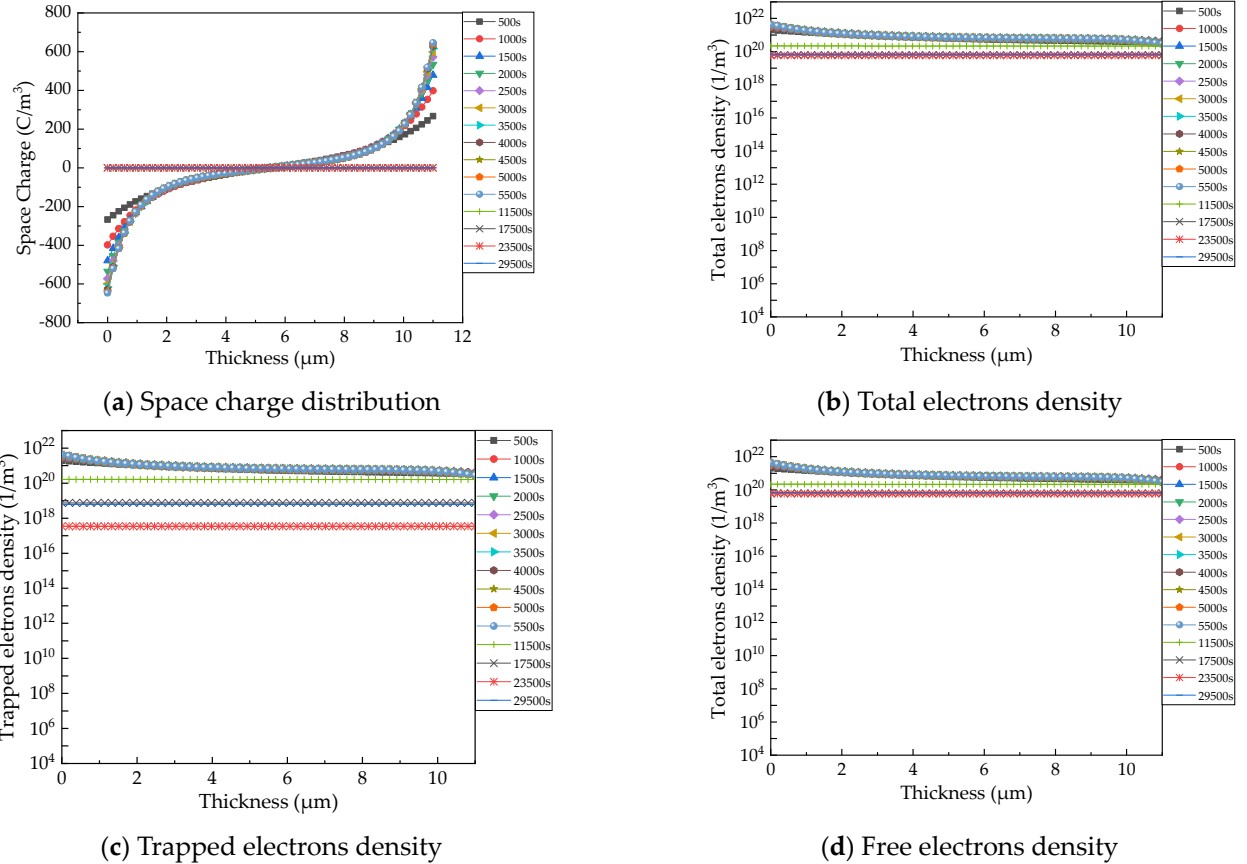

**(a)** Space charge distribution

**(b)** Total electrons density

**(c)** Trapped electrons density

**(d)** Free electrons density

**Figure 7.** Simulation results of space charge distribution (**a**) and electrons density distribution (**b**–**d**) at 60 °C (0 s–6000 s: 230 kV/mm; 6000 s–12,000 s: 270 kV/mm; 12,000 s–18,000 s: 310 kV/mm; 18,000 s–24,000 s: 350 kV/mm; 24,000 s–30,000 s: 390 kV/mm. When the electric field is greater than 230 kV/mm, space charge distribution and electrons density distribution are almost the same at each time).

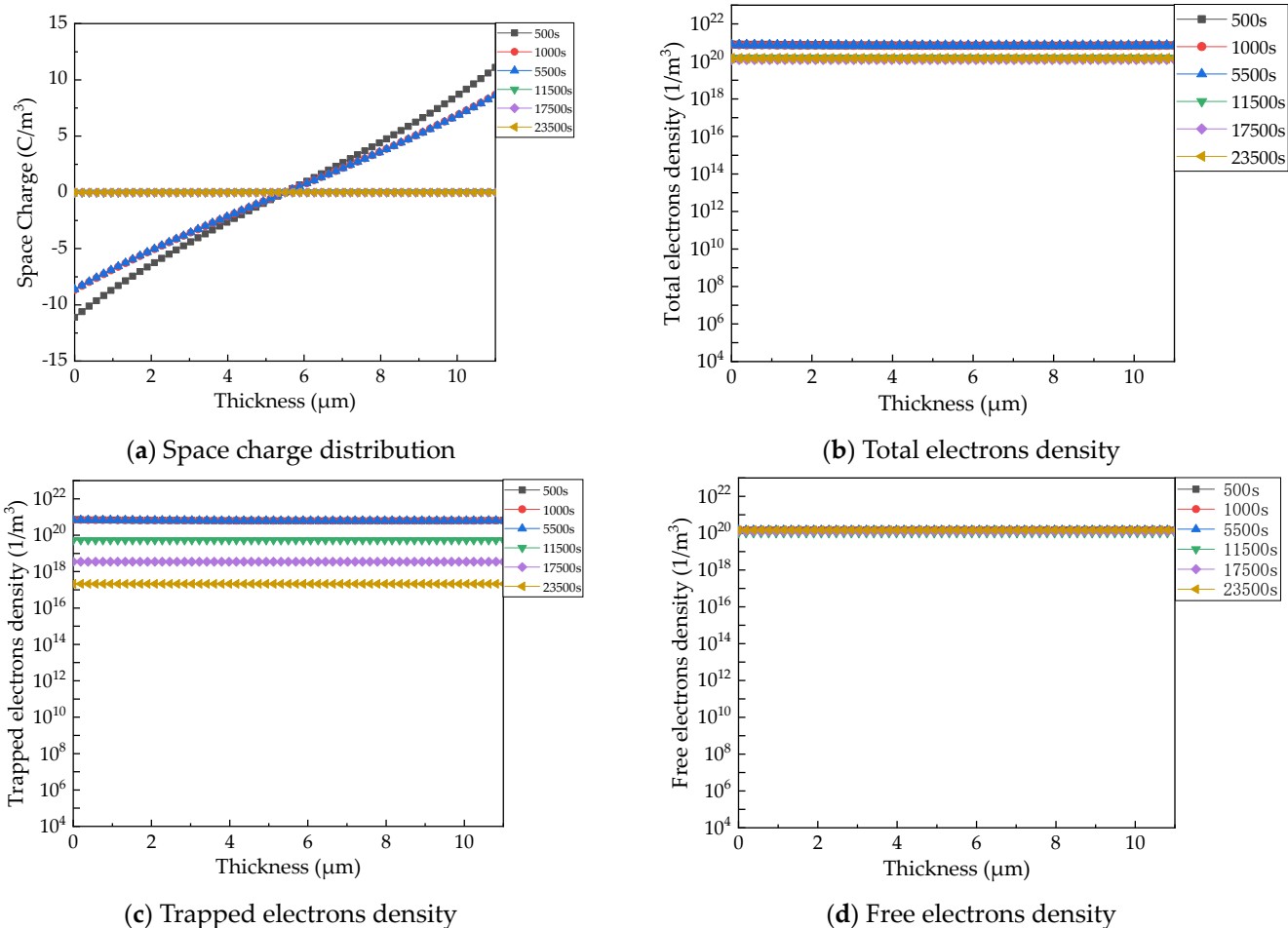

**Figure 8.** Simulation results of space charge distribution (**a**) and electrons density distribution (**b**–**d**) at 80 °C. (0 s–6000 s: 230 kV/mm; 6000 s–12,000 s: 270 kV/mm; 12,000 s–18,000 s: 310 kV/mm; 18,000 s–24,000 s. When the electric field is greater than 230 kV/mm, space charge distribution and electrons density distribution are almost the same at each time.).

The transient process of charge trapping and space charge formation is much more rapid, and space charge accumulation is much less at 60 °C and 80 °C, in contrast to that at 20 °C and 40 °C, as shown in Figures 7 and 8. This agrees well with the fact that the slope of the SCLC curve at 60 °C and 80 °C is much less than that at 20 °C and 40 °C, as shown in Figure 3. Therefore, charge transport simulation is helpful to provide deep insights into the space charge transient process.

### 3.2. Experimental and Simulation Results on DC Electrical Breakdown

Since the electrical breakdown of insulating materials generally has great dispersion and uncertainty, the two-parameter Weibull distribution was used to conduct statistical analysis on the breakdown test data. Ten samples were tested at each temperature and the electrical breakdown strength $E_b$ was fitted to the Weibull distribution, as shown in Figure 9, and the Weibull parameters are shown in Table 3. The scale parameter $\alpha$ is the characteristic electrical breakdown strength, with a failure probability of 63.2%. The shape parameter $\beta$ characterizes the dispersion of the electrical breakdown strength $E_b$ and the smaller value of $\beta$ implies less dispersion of $E_b$. Little changes (less than 3%) can be found in the electrical breakdown strength for the temperature varying from 20 °C to 60 °C. It is unusual to find that $E_b$ at 20 °C is a little lower than that at 40 °C. To confirm the fact, the $E_b$ of additional ten samples are tested at 20 °C, and the results are shown together in Figure 9. The electrical breakdown strength $E_b$ at 80 °C is about 10% lower than that at 60 °C.

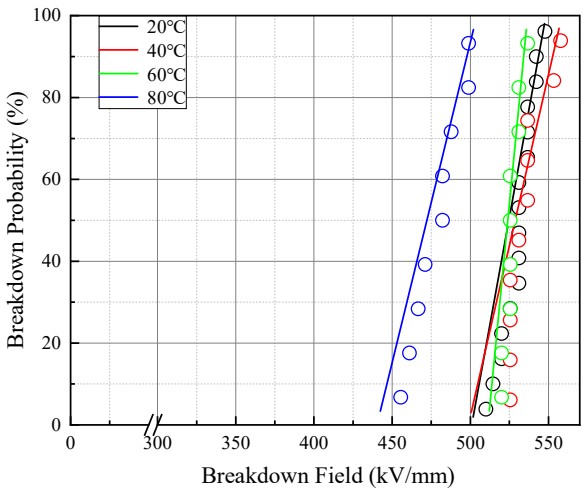

**Figure 9.** Breakdown behavior of PP films at different temperatures.

**Table 3.** Parameters for breakdown behavior.

| Temperature | 20 °C | 40 °C | 60 °C | 80° C |
|---|---|---|---|---|
| $\alpha$(kV/mm) | 535.23 | 541.30 | 529.40 | 485.42 |
| $\beta$ | 61.11 | 44.41 | 101.59 | 36.36 |

Figure 10 shows the simulation on space charge distribution in BOPP films, under a linearly increasing electric field with time, at different temperatures.

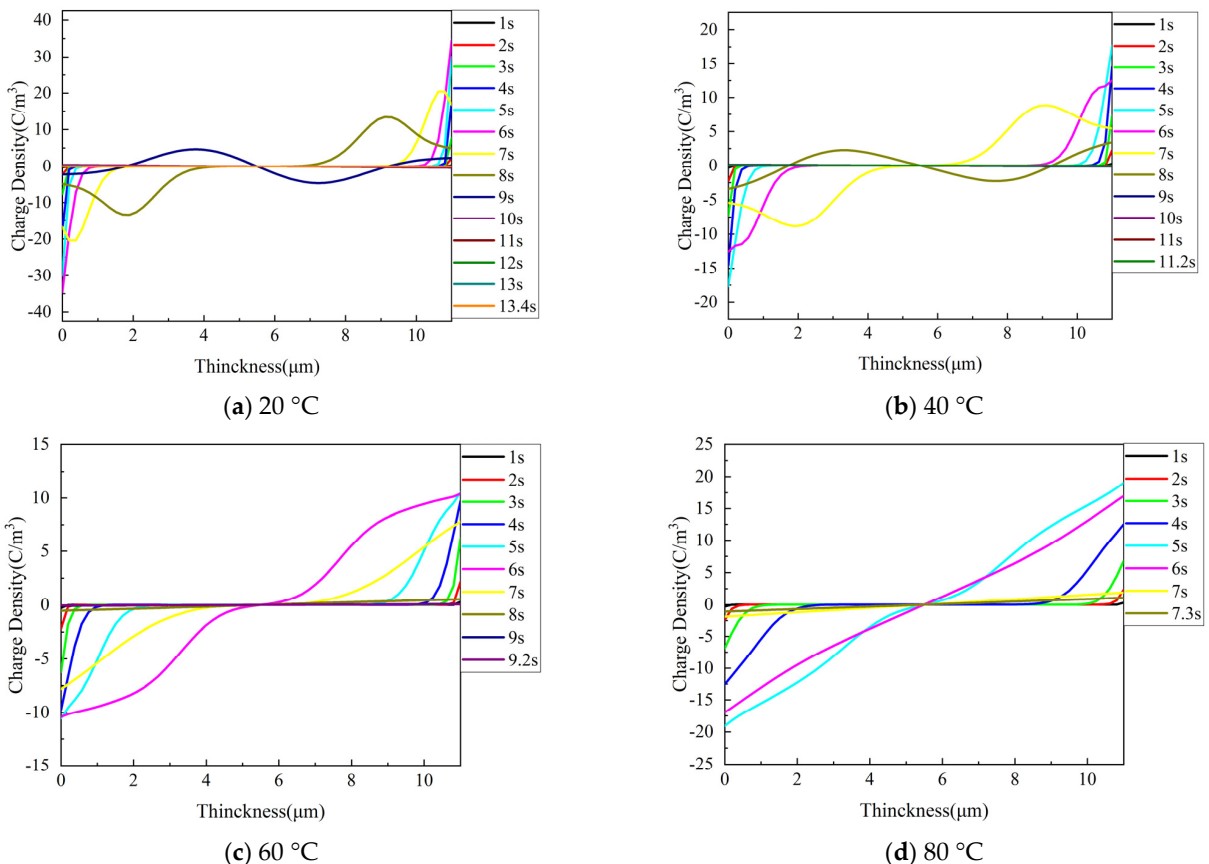

**Figure 10.** Simulation on space charge distribution in BOPP films under linearly increasing electric field with time at different temperatures.

This study investigates the experimental measurements and comparison of the breakdown voltage of experiments and simulations. With the comparison of experimental results, the selection of parameters for the simulation model on DC breakdown strengths is credible. The simulation model contributes to understanding the underlying mechanism of how the space charge influences the breakdown process.

As shown in Figure 10a–c, when the electric field is applied for 2 s, a small amount of the homo space charge appeared on both sides of the insulation. Then, the positive and negative charges injected on both sides increase gradually and migrate inwards, the space charge packet occurred at 7 s, 6 s, and 5 s, respectively, and the negative charge moved from the cathode to the anode, and the positive charge moved from the anode to the cathode at the same time. The amplitude of the space charge packet in the moving decreases gradually. The hetero space charge formed near both the electrodes, due to the migration of carriers Finally, the overall space charge was near to zero, until the film broke down. In Figure 10d, carriers migrate faster due to the higher temperature, so the electric field did not reach the threshold field that initiated the charge packets [16]. At 20–80 °C, the maximum homo space charge accumulated on both sides is 35 C/m$^3$, 18 C/m$^3$, 11 C/m$^3$, 18 C/m$^3$, respectively.

We present the results of the electrical conductivity and breakdown of PP as a function of temperature and field strength, to study charge transport under DC stresses.

The transient currents of PP are manifestations of polarization, which have been related to the molecular mechanism that is responsible for the glass transition. We worked with temperatures considerably above glass transformation temperature, and, indeed, found that transients could essentially be traced to geometric, rather than molecular, origins. High fields influence both the concentration and mobility of the space charge. A polymer is a long-chain macromolecule, formed by the polymerization of small molecular monomers. The single bond in the molecule can rotate internally, so that the molecule has many different conformations. Due to the thermal movement, the molecular conformation keeps changing, and the molecular size will also change [17].

With the increase in temperature, the thermal expansion of polypropylene occurs and its volume keeps increasing. The lattice structure becomes a little bit bigger, and the amorphous region becomes looser, which contributes to structural inhomogeneities and free volume in the film, and can thus directly influences the dielectric strength of the film specimen [18,19]. Breakdown is initiated by free electrons that are accelerated by the electric field in the largest of the holes in the amorphous phase, according to the free-volume theory. With the increase in temperature and electric field, the free electrons move faster, the conductive current gets higher and higher. When the energy that is gained by free electrons attains the barrier energy with the probability P = 1, the breakdown occurs.

Because the microvoids may act as deep traps and enhance the breakdown strength, the structure of the small crystal size and dense amorphous part may have a higher breakdown strength [20]. Deep traps can form a space charge capture center, which is easier to cause accumulation of the space charge. Moreover, a larger void volume means that the original trap depth becomes shallow, and the retention time of the carriers in more shallow depths becomes shorter. It shows that the shallow trap has little hindrance to the carriers, and the carriers can trap and detrap more easily, and the migration distance of the carriers is longer at the same time. So, shallow traps can assist carrier conduction in the medium. The increase in the applied electric field, temperature, and shallow traps will accelerate the transport speed of the charge carriers, so that the carriers are not easy to be captured by the traps. Electrical breakdown is dependent on the microvoids size and their distribution in size. On the basis of the feature of the morphology and electronic structure of PP, the energy for electrons to break chemical bonds in the electrical breakdown is the band gap of the polymer instead of the bond energy. Electronic state transition and voids in the polymer are two necessary conditions of electrical breakdown.

The simulation results of the conductive currents show that the distribution of homo space charge is dispersive along the thickness direction of the sample injected from two

electrodes. For 20 °C and 40 °C, more homopolar charges accumulate near the poles, which gradually migrate to the middle of the sample after 35,500 s and 11,500 s, respectively. As positive and negative charges migrate to the middle of the sample, the space charge density drops to almost zero, and the electron density distribution becomes uniform. However, at 60 °C and 80 °C, the accumulation of the space charge at the electrodes is less, and it migrates to the middle of the sample at 11,500 s and 5500 s, respectively. The space charge density drops to almost zero, and the electron density is almost uniform. When the initial homo space charge density is higher, it takes a longer time for the electrons to reach equilibrium. With the increase in the temperature, carriers in the polypropylene migrate faster and faster, leading to less space charge accumulation, and the faster speed space charge distribution and electronic distribution reach equilibrium. The total electron density and free electron density increased with the temperature before the breakdown, and the detrapped electron density decreased with the temperature [21]. At 20–60 °C, with the increase in the electric field, the total electrons density and free electrons density firstly reach uniform distribution with the increase in time, and then slowly decrease until the breakdown. The trapped electrons density also reaches uniform distribution with the increase in time, and then slowly decreases, and finally increases to varying degrees before the breakdown. At 80 °C, with the increase in the electric field, the electrons density immediately reaches the uniform distribution state, and the free electrons density almost remains unchanged, while the total electrons density and trapped electrons density slowly decrease until the breakdown. The effect of temperature and electric field on the free charge density will affect the dielectric conductivity. The injected charge can gain more energy at a higher temperature and electric field, which increased the conductivity of the polypropylene film. The greater the conductivity current, the more easy to break down.

The impact of hot electrons on molecular segments is a prerequisite for breakdown. Several models, such as collective breakdown, thermal breakdown, free volume breakdown, electromechanical breakdown, and secondary effects, have been proposed to interpret the breakdown phenomena in polymeric insulating materials. The actual breakdown mechanism will not be a single model. Under the high electric field, the speed and density of the electrons and holes are very large. Space charge accumulation caused the energy gain in free volumes and the distortion of the electric field. The energy carriers gained is related to the local electric field. When the electron energy is large enough, it will travel across the potential barrier, causing a large local current. The large local current will heat the local point of the insulating material in a short time, causing breakdown.

The breakdown strength of polypropylene possesses temperature dependence to some extent [22]. With the increase in temperature, the breakdown field strength of polypropylene remains unchanged, or decreases slightly, and decreases obviously after reaching a certain temperature. The decrease in breakdown field strength is due to the increase in free volume. The glass transformation temperature of PP is below 0 °C, and the free volume of PP changes greatly when the temperature is above the glass transformation temperature [23].There is no temperature dependence below 80 °C, and the sudden drop in breakdown field strength at 80 °C suggests two possible breakdown mechanisms. At 20–60 °C, the breakdown strength is almost independent of the temperature and applied voltage, and we assumed that the breakdown was caused by electronic processes. At 80 °C, the breakdown mechanism is the interaction of thermal breakdown and electronic processes. It shows that even a short run at a high temperature will cause great damage to the insulation of the polypropylene film [24].

By investigating the space charge change during the process of the breakdown, we found that the homo space charges injected by the electrodes migrate to the inside of the material, and finally to the opposite electrode. The homo space charges become the hetero polar charges near the opposite electrodes, resulting in the local high electric field near the positive and negative electrodes. At 20–60 °C, the space charge packet is excited. When the high electric field distortion near the electrode reaches the breakdown threshold, the breakdown of the polypropylene film will occur. The charge density approaching zero is

the phenomenon of hetero space charge formation, which is also the precursor of electric breakdown [25]. The change in space charge density in the simulation of the breakdown process is similar to the change in charge density in the simulation of conductance currents, of which the breakdown mechanism will not be described here.

## 4. Conclusions

We have studied the charge transport in a polypropylene film during the progress of conductivity and breakdown. The following conclusions can be drawn by analyzing the experimental and simulation data:

(1) With the increase in temperature, the molecular conformation of PP keeps changing, the free volume of PP increases, which facilitates charge injection and transport. Thermal stress plays a greater role compared to the electric field in charge transport at both short-term and long-term high temperatures. The increase in free electrons density at high temperatures improves the conductivity currents and breakdown probability of polypropylene. The increase in conductivity currents is the feature.

(2) In the process of short-time breakdown, the breakdown mechanism is different at different temperatures. At 20–60 °C, electron breakdown acted, and at 80 °C, thermal breakdown and electron breakdown acted together. The breakdown strength decreased significantly, indicating that the short-time thermal environment will cause great damage to the insulation of the polypropylene film. The charge transport process in breakdown simulation is basically similar to that in conduction current simulation.

(3) In the process of charge transport, the space charge density dropped to close to zero, which was a sign that the electrons and holes inside the polypropylene film reach a stable state. The overall electric field was zero, but the carrier transport speed was very fast, which still causes the continuous increase in conduction currents. There would also be a high local electric field to cause the breakdown.

**Author Contributions:** F.T. conceived the presented idea and analyzed the research data. The article is originally written by F.T. and revised by author S.Z., X.D., F.L., Z.X. and J.W. All authors have read and agreed to the published version of the manuscript.

**Funding:** This study is supported by the State Key Laboratory of Advanced Power Transmission Technol-ogy (grant No. GEIRI-SKL-2019-006).

**Acknowledgments:** This study is supported by the State Key Laboratory of Advanced Power Transmission Technology (grant No. GEIRI-SKL-2019-006).

**Conflicts of Interest:** The authors declare no conflict of interest.

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
