# Peer review of "Investigation on Charge Transport in Polypropylene Film under High Electric Field by Experiments and Simulation"

_energies, doi:10.3390/en14164722_

Round 1

Reviewer 1 Report

In this paper, the authors investigate the charge transport behavior and the breakdown of commercial biaxial-orientated polypropylene films under different temperatures by both experiments and simulation.

The topic is important from both a fundamental and an application standpoint as polypropylene films are used as insulation in HVDC filter capacitors.

The paper is clear and well organized. It can be published after a minor revision.

1. “In the breakdown process at different temperatures, polypropylene film is mainly electric breakdown process at 20-60°C, and the breakdown strength is almost unchanged.” The sentence is unclear. Rephrase it.

2. Figure 2: plot (c) – modify the y-axis scale to be consistent with the format used for the other plots of the figure.

3. “The lnI changes nearly linearly with lnE at high electric fields with 164 the slope of the curve higher than 1, which is the typical characteristics of the space charge limited conduction current (SCLC) behavior[13].” The authors could note here that different conduction mechanisms have been reported in polypropylene such as Schottky emission and ionic transport. See for instance https://doi.org/10.1088/0022-3727/42/13/135405 that can be added to the reference list.

4. “This indicates that the conduction current is generally dominated by SCLC process, i.e. by charge accumulation, charge trapping and detrapping process.” The authors could elaborate a bit more here. Which is the nature and the origin of the traps they are considering?

5. Correct the typo: “space charge packet ocuurred at 7 s”

6. “Because the tiny nano-pores may act as deep traps and enhance the breakdown strength, the structure of small crystal size and dense amorphous part may have higher breakdown strength[19].” I wonder if the authors have any evidence of the nano-pores here mentioned.

Reviewer 2 Report

The manuscript is clearly written with only a few typos and provides information that can be useful for researchers in the field.

There are however two main points that need to be further developed:

  1. Please provide a better review of the state of the art. Polypropylene films under high electric fields have been previously investigated. Authors should state what is the progress beyond the state of the art of their manuscript related to what is already known.
  2. Figure 2 shows current curves following different patterns depending on electric field levels... first decreasing and next decreasing... always decreasing, always increasing. What is the origin of this apparent random behavior between temperatures and a given temperature when varying the electric field?

Additional question: What are the humidity condition of the experiments. 60ºC drying is established in the preparation process but no RH information during the experiment is provided.

Reviewer 3 Report

This manuscript reports on a joint experimental/modeling study of the behavior of bioriented polypropylene under very high electrical stress. This is a major question in the context of the use of such insulating materials in capacitors. The work is well-positioned with respect of the state of the art, the research strategy is clear and solid, the presentation of the results is well-organized and a detailed interpretation of the electrical breakdown mechanism is proposed. Overall, the manuscript brings significant new insight in the field and deserves to be published. Only a couple of minor/technical points should be corrected:

  • The expression ‘traps fulfilled’ on line 174 should probably read ‘traps fully filled’.
  • It is mentioned on line 188 that the current decreases with increasing temperature. Actually the current increases with the temperature (see fig. 4), as expected for thermally-activated transport.
  • The manuscript is ‘data-intensive’, with many graphs and figures. Probably some figures could be moved to the SI section (for instance, among the Fig. 5 to Fig. 8 sequence).
  • To support the discussion of page 13, it would be helpful to provide the reader with the degree of crystallinity of the BOPP used in the study.
  • What the authors call ‘glassy transformation temperature’ is more commonly called ‘glass transition temperature’.

To conclude, I recommend acceptance of this manuscript for publication in Energies once these minor points have been addressed.

Round 2

Reviewer 2 Report

No further comments

This manuscript is a resubmission of an earlier submission. The following is a list of the peer review reports and author responses from that submission.